Using multi-scale distribution and movement effects along a montane highway to identify optimal crossing locations for a large-bodied mammal community

Schuster Richard 1 2 mail@richard-schuster.com
Römer Heinrich 1
Germain Ryan R. 2
1 Neurobiology and Behavior Group, Department of Zoology, Karl-Franzens-University , Graz , Austria
2 Centre for Applied Conservation Research, Department of Forest & Conservation Sciences, University of British Columbia , Vancouver, BC , Canada
Stewart Gavin
Electronic publication date: 2013 Oct 24
Publication date: 2013
Volume: 1
Electronic Location ID: e189
Received 2013 Jul 30; Accepted 2013 Oct 8
Copyright: © 2013 Schuster et al.
Copyright year: 2013
Copyright holder: Schuster et al.
License: This is an open access article distributed under the terms of the Creative Commons Attribution License, which permits unrestricted use, distribution, and reproduction in any medium, provided the original author and source are credited.
License URL: https://creativecommons.org/licenses/by/3.0/

Keywords: Connectivity, Mitigation, Multi-species, Habitat fragmentation, Snow-tracking, Wildlife

Funding: Federal Ministry of Science and Research, Austria Karl-Franzens-University University of British Columbia This project was supported by a masters student grant to RS from the Federal Ministry of Science and Research, Austria. Institutional support was provided by the Karl-Franzens-University (RS, HR) and the University of British Columbia (RS, RRG). The funders had no role in study design, data collection and analysis, decision to publish, or preparation of the manuscript.

==============================
Roads are a major cause of habitat fragmentation that can negatively affect many mammal populations. Mitigation measures such as crossing structures are a proposed method to reduce the negative effects of roads on wildlife, but the best methods for determining where such structures should be implemented, and how their effects might differ between species in mammal communities is largely unknown. We investigated the effects of a major highway through south-eastern British Columbia, Canada on several mammal species to determine how the highway may act as a barrier to animal movement, and how species may differ in their crossing-area preferences. We collected track data of eight mammal species across two winters, along both the highway and pre-marked transects, and used a multi-scale modeling approach to determine the scale at which habitat characteristics best predicted preferred crossing sites for each species. We found evidence for a severe barrier effect on all investigated species. Freely-available remotely-sensed habitat landscape data were better than more costly, manually-digitized microhabitat maps in supporting models that identified preferred crossing sites; however, models using both types of data were better yet. Further, in 6 of 8 cases models which incorporated multiple spatial scales were better at predicting preferred crossing sites than models utilizing any single scale. While each species differed in terms of the landscape variables associated with preferred/avoided crossing sites, we used a multi-model inference approach to identify locations along the highway where crossing structures may benefit all of the species considered. By specifically incorporating both highway and off-highway data and predictions we were able to show that landscape context plays an important role for maximizing mitigation measurement efficiency. Our results further highlight the need for mitigation measures along major highways to improve connectivity between mammal populations, and illustrate how multi-scale data can be used to identify preferred crossing sites for different species within a mammal community.

Introduction

As human-induced fragmentation of wildlife habitats continues to increase, there is a growing need to both investigate the effects of such fragmentation on animal communities and to present possible solutions to help mitigate these effects (e.g., Gonzalez et al., 1998; Crooks, 2002). Roads are a major contributor to the fragmentation of wildlife habitat around the world (e.g., North America: Trombulak & Frissell, 2000; Underhill & Angold, 2000; Europe: Holderegger & Di Giulio, 2010; Selva et al., 2011; Australia: Jones, 2000), and their construction and maintenance are one of the most widespread forms of human-based habitat modification (Bennett, 1991; Noss & Cooperrider, 1994). Major effects of roads on wildlife can include traffic mortality, modification of animal behavior (e.g., road avoidance), and alteration of the physical and chemical environment leading to barrier effects and habitat fragmentation (reviewed in Trombulak & Frissell, 2000; Jaeger et al., 2005). Movement barriers such as roads can affect wildlife at several different levels; in addition to lowering individual fitness through restricted access to resources and increased mortality risk (reviewed in Fahrig & Rytwinski, 2009), roads may also reduce gene flow between fragmented habitats and contribute to the creation of smaller subpopulations which are more vulnerable to stochastic events (Boyce, 1992; Forman & Alexander, 1998; Jaeger et al., 2005). For example, road fragmentation is implicated as a major contributor towards the extirpation of carnivorous mammals in the Rocky Mountains of western North America (Noss et al., 1996). Thus, there is a clear need for research on predicting areas of preferred animal crossing sites to both identify appropriate locations for mitigation measures and help reduce the negative effects of roads on wildlife communities.

Most studies investigating how to apply practical mitigation measures (e.g., crossing structures such as overpasses) aimed at reducing the effects of roads on animal communities focus on predicting the landscape features of animal-vehicle collision sites (e.g., Malo, Suarez & Diez, 2004; Seiler, 2005). Although current funding for mitigation measures is often allocated to sites along roads where collisions have previously been reported, collision sites may not necessarily represent the areas preferentially used by wildlife to cross roads (Alexander, Waters & Paquet, 2005). Consequently, identifying the landscape features surrounding roads which represent both preferred and avoided animal crossing sites may help inform mitigation design and optimize animal movement between sub-populations, thereby reducing the effects of habitat fragmentation (Singleton & Lehmkuhl, 1999; Alexander, Waters & Paquet, 2005).

Previous studies on the efficiency of mitigation strategies indicate that different mammal species can be highly variable in their tolerance to human structures, suggesting that the effects of barriers such as roads and the success of mitigation strategies will also likely vary by species (Beier & Noss, 1998; Trombulak & Frissell, 2000). Studies investigating mitigation strategies for high-traffic areas should therefore incorporate multiple focal species and predict spatial linkages across roads at the community level (Beier, Majka & Spencer, 2008). In particular, modeling animal movement across multiple spatial scales may aid our understanding of preferred habitat use along roads when considering multiple species of large mammals, which may each differ in terms of habitat requirements, home range sizes, and sensitivity to road disturbance. Animals may also select movement habitat at multiple scales, as shown in migratory birds, reptiles, and large mammals (e.g., Boyce et al., 2003; Beaudry, deMaynadier & Hunter, 2008; McClure, Rolek & Hill, 2012). Therefore, studies which incorporate several spatial scales into the same analytical framework, and compare results of predicted crossing sites across multiple spatial scales may prove particularly useful in planning mitigation strategies. Because micro-habitat assessments are often costly and labor-intensive (e.g., Fearer et al., 2007), direct comparisons of the validity of predictive models generated from micro-habitat data versus macro-habitat assessments from remotely sensed data may aid future research in allocating more time and funding to the most efficient methods.

Here, we characterize preferred and avoided crossing sites of eight large-bodied mammal species along a 95 km length of highway through the Purcell Mountain Range of North America. We use a multi-scale approach comparing high-resolution, manually-digitized habitat metrics with remote sensing-derived metrics at three spatial scales (200 m, 500 m, and 1 km) to investigate the potential drawbacks of each method in implementing mitigation measures. Our goals for this study include identifying the habitat variables (‘predictors’) of preferred and avoided crossing sites for each mammal species along this highway, and evaluating the efficiency of using macro-habitat predictors derived from freely available remote sensing data versus manually-digitized micro-habitat maps to predict such crossing sites. To address these goals we ask the following specific questions: (1) does the highway present a movement barrier to a multi-species community of mammals, (2) do species show preference in their choice of crossing sites towards predefined landscape predictors, (3) are there preferred crossing areas for species or species groups along the highway that could potentially serve as mitigation sites, (4) are preferred versus avoided crossing sites better predicted by habitat variables generated at the macro-scale, micro-scale, or a combination of both?

Methods

Study area

Our study was conducted along Southern Trans-Provincial Highway 3 (hereafter Hwy 3) between the towns of Creston and Cranbrook, in south-eastern British Columbia, Canada (Fig. 1). The study area is located in the Purcell Mountain Range, which ranges from 620 m to 2,087 m in elevation, and is comprised of Interior Cedar Hemlock and Interior Douglas Fir Biogeoclimatic zones (Meidinger & Pojar, 1991). We chose this study area for its ecological importance as a trans-boundary priority area (Yellowstone to Yukon Conservation Initiative, 2013) that connects small populations of carnivores such as grizzly bears (Ursus arctos horribilis) and Canada lynx (Lynx canadensis) along the Canada – USA border. Hwy 3 bisects this important corridor, possibly leading to negative effects on the connectivity of this movement corridor for mammal populations. The average annual traffic volume (AADT) for this highway section was 3050 cars/day in 2007, with a seasonal (December to March) average of 2020 vehicles/day (British Columbia Ministry of Transportation and Infrastructure, 2010).

Figure 1 Study Area (Cranbrook 49° 30′ N, 115° 46′ W).

East Kooteney region, South eastern British Columbia, Canada. Also shown are the data collection points as well as the remote sensed (EOSD) class distribution that formed part of the model inputs.

Data collection

We monitored species movement through the study area by recording tracks in the snow where animals attempted to cross Hwy 3, as well as along ten transects approaching the highway, set back from any highway right-of-way (distance from transects to highway ranged from 10 to 900 m, mean 175 m). We pre-defined our transects as survey lines marked with flagging tape, roughly parallel to the highway. Highway and transect tracks were recorded over two winter seasons, January to March 2007 and December 2007 to February 2008 (all observations recorded by RS).

Highway and transect track surveys were conducted using methods similar to Van Dyke, Brocke & Shaw (1986), and Alexander, Waters & Paquet (2005). Briefly, we conducted highway crossing attempt surveys along a 95 km length of Hwy 3, at least 12 h after the last snowfall. Each survey was conducted from a moving vehicle with a speed of approximately 10–15 km/h. When a track was observed, the investigator stopped the vehicle and conducted an on-foot inspection to identify the track. In total, we investigated tracks for 12 mammal species: coyote (Canis latrans), fox (Vulpes vulpes), wolf (Canis lupus), cougar (Puma concolor), bobcat (Lynx rufus), lynx, marten (Martes americana), wolverine (Gulo gulo), elk (Cervus canadensis), moose (Alces alces), white-tailed and mule deer (Odocoileus virginianus and Odocoileus hemionus, respectively). When we were uncertain of the identity of a track, we recorded track pattern measurements, took photos and later consulted field guides (Sheldon, 1997; Elbroch, 2003) for identification. Data at a total of 463 crossing sites were georeferenced with a handheld, Garmin eTrex Summit GPS receiver (WGS 1984, ±10–40 m). If multiple tracks were found for one species at a single crossing area, we recorded the total track count. We also recorded the success of a crossing attempt, here defined as the presence of a continuing set of tracks on the opposite side of the road. When tracks of the same species were found within 300 m of a crossing site, it was not recorded as an individual crossing attempt, but rather as a possible repeat crossing of the same individual (Alexander, Waters & Paquet, 2005). Surveys were suspended when continuous heavy snowfall covered tracks during data collection.

Transects were established off-road in suitable areas close to the highway. Suitability was contingent upon minimal disturbance from residential areas, and no barriers to observer access (i.e., lakes, steep terrain, fences or private property). Seven transects had a linear distance of 1 km, while one was 2 km (Transect 6) and one was 5.4 km in length (Transect 10). Only the first kilometer of transect 10 was surveyed during the second season of data collection, and this was classified as Transect 9 for ease of data handling. We recorded tracks of the same species according to the protocol of the crossing attempt surveys, and georeferenced a total of 308 individual track locations along the transects. We surveyed transects between 12 and 96 h after snowfall, usually starting the day following a road survey, with 5 to 7 km of transect being surveyed per day. Due to the limited number of tracks recorded for carnivores (coyote, bobcat, cougar, wolf, fox, lynx, marten, wolverine, see Results) we grouped all the above species into one category ‘carnivores’, while evaluating the remaining species of ‘ungulates’ (moose, elk, deer) separately for landscape variable preference models and predictive mapping, and both separately/combined for estimates of permeability across the highway.

Road barrier effect

We standardized the highway crossing attempt and transect survey data by the number of 12 h periods that had elapsed since the time of the last snowfall to correct for time effects (Thompson et al., 1989). For calculation of the road barrier effect, we standardized survey data for the highway and transects by kilometers surveyed: Crossings per km=Total number of tracksTotal length of surveys

We then calculated the permeability of the highway by standardizing the crossings per km of highway with the crossings per km of transect: Permeability=Highway crossings per kmTransect crossings per km

We also constructed track accumulation curves along the 95 km of highway for all four species groups to identify areas of the highway with greater crossing intensity for each mammal group.

Multi-scale landscape variables

To develop our micro-habitat assessments, we imported the collected GPS data into ArcGIS 9.3 (ESRI, 2009). The GPS points from the highway surveys and the transect surveys were set on top of a georeferenced (WGS 1984, UTM Zone 11N) orthophotograph layer from 2004, with a spatial resolution of 1 m, provided by a Web Map Service (WMS) of GeoBC (http://www.geobc.gov.bc.ca). For each GPS point, we created a circular buffer of 200 m to represent the perceptual area of the animal directly influenced by the surrounding landscape predictors (e.g., Lingle & Wilson, 2001), which we define as ‘perceptual area polygon’. For each buffer area, we digitalized polygons for predefined landscape predictors and georeferenced them using the orthophotograph layer and Google Earth, as the latter provided more recent images of the research area. We used the following landscape predictors, adapted from Dickson, Jenness & Beier (2005): forested (forest + woodland), shrub, herbaceous (grassland + agriculture), riparian, water, non-vegetated (gravel, rock + dirt), highway ( + shoulder), road/path, railroad, residential, developed, disturbed and wetland (Table 1). We then calculated the percentage of each buffer area overlapped by each landscape predictor.

Table 1 Predictor variable description.

Variables used in models predicting preferred and avoided crossing sites at 200 m, 500 m, and 1 km spatial scales. Perceptual area polygons were only recorded at the 200 m scale and variables were hand-digitized from 1-m pixel photos.

Variable name	Variable description	Source	
Forest	forested (forest + woodland)	Perceptual area polygon
Variable units: % of 200 m radius area	
Shrub	shrub	
Herb	herbaceous (grassland + agriculture)	
Riparian	riparian	
Freshwater	river + lake	
Unvegetated	non-vegetated (gravel, rock + dirt)	
Highway	highway ( + shoulder)	
Road	road/path	
Railroad	railroad	
Residential	residential + developed	
Disturbed	disturbed habitat (e.g., excavation sites)	
Wetland PAP	wetland	
Water	Lakes, reservoirs, rivers, streams, or salt water.	EOSD
Variable units: area [m2] in spatial scale buffer around data point	
Exposed	River sediments, exposed soils, pond or lake sediments, reservoir margins, beaches, landings, burned areas, road surfaces, mudflat sediments, cutbanks, moraines, gravel pits, tailings, railway surfaces, buildings and parking, or other non-vegetated surfaces.	
Low shrub	At least 20% ground cover which is at least one-third shrub; average shrub height less than 2 m.	
Wetland	Land with a water table near/at/above soil surface for enough time to promote wetland or aquatic processes; Trees + Shrub + Herb	
Herbecous	Vascular plant without woody stem (grasses, crops, forbs, gramminoids); minimum of 20% ground cover or one-third of total vegetation must be herb.	
Dense conifer forest	Greater than 60% crown closure; coniferous trees are 75% or more of total basal area.	
Open conifer forest	26–60% crown closure; coniferous trees are 75% or more of total basal area.	
Open broadleaf forest	26–60% crown closure; broadleaf trees are 75% or more of total basal area.	
Gravel road length	Road length within buffer (gravel) [m]	TRIM	
Paved road length	Road length within buffer (paved) [m]	
Buildings	Number of buildings within buffer	

Because large mammals might respond to both fine and coarse scale habitat features (e.g., Mayor et al., 2007), we developed a series of variables describing macro-habitat landscape features at three spatial scales: 200 m, 500 m, and 1 km. For modeling species abundance along the highway and transects, we chose candidate predictor variables based on their ability to predict species abundance at site and landscape levels in similar studies (e.g., Malo, Suarez & Diez, 2004; Guisan & Thuiller, 2005). All remotely sensed predictors (Table 1) were derived from the following sources: Terrain Resource Information Management (TRIM, Province of BC 1992) and Earth Observation for Sustainable Development Landcover (EOSD LC 2000, Wulder et al., 2008). Our dataset comprised 12 predictor variables from the perceptual area polygons and 11 from remote sensing on 3 scales (200 m, 500 m, 1 km; Table 1), derived at each of 463 highway locations and 308 transect locations. All remote sensing predictors were created using Geospatial Modelling Environment (Beyer, 2012) in conjunction with ArcGIS 10 (ESRI, 2010) and R v. 2.15.2 (R Development Core Team, 2012). Due to their widely varying scales, all predictors were standardized to mean = 0, sd = 1 to ensure that their importance was not driven by measurement scale (White & Burnham, 1999).

Landscape variable preference models

Since predetermining the appropriate data distribution for our count data from ecological knowledge alone was not possible, we modeled abundance incorporating both a Poisson distribution (P) and negative binomial (NB) distribution to account for potential overdispersion (e.g., Zeileis, Kleiber & Jackman, 2008). Because of the large proportion of zero values included in our data-set, we also applied zero-inflated models (ZIP, ZINB; Lambert, 1992), which are mixture models that combine both count data and a binomial model. To determine which of these distributions best represented our species data, we visually inspected the data and compared the log likelihood, AIC, and number of correctly predicted zeros for each distribution model fits using intercept-only models. To test for differences among distribution functions, we used likelihood ratio tests to compare the Poisson and negative binomial distributions, since the Poisson distribution is a restriction of the more general negative binomial distribution (Hilbe, 2008). We tested H0 for no difference between the two and H1 that the negative binomial was a better fit to the data. We tested the same hypothesis using the zero-inflated Poisson and zero-inflated negative binomial. Next, we used a Vuong test (Vuong, 1989; Greene, 1994) to evaluate whether the zero-inflated models were a statistically better fit to the data than their base model (Hilbe, 2008). The Vuong test is generally formulated as: V=sqrtN*meanmsm*mm=lnμ1μ2

where μ1 = predicted probability of y for the zero-inflated model, μ2 = predicted probability of y for the base model, sm = standard deviation of m, and N = number of observations in each model, where both must use the same observations. The test statistic V is asymptotically normal. If V > 1.96, the zero-inflated model is preferred; if V < −1.96, the base model is preferred; and if the value of V is between −1.96 and 1.96 neither model is preferred (Hilbe, 2008). To perform these tests we used the function vuong in R package pscl v.1.04.4 (Zeileis, Kleiber & Jackman, 2008; Jackman, 2012).

We compared our measures of model selection (AIC, LogLik, predicted zeros, Vuong) for all four distributions (P, NB, ZIP, ZINB) throughout each of the model building stages of this study to avoid bias in predetermining the distribution with intercept-only models. For each of the four mammal groups (deer, elk, moose, carnivores), we compared models of predicted habitat preference using each distribution (P, NB, ZIP, ZINB) and data set (Highway, Transect) across six separate spatial approaches: (i) 200 m scale, (ii) 500 m scale, (iii) 1 km scale, (iv) all 3 scales combined; (v) perceptual area polygons; (vi) all scales and perceptual area polygons combined. For spatial approaches (iv) and (vi), we created an iterative model fitting procedure that starts with an intercept only base model, individually adds predictor variables, records the results of each fitted model, and retains all top-ranked models (ΔAICc ≤ 2) at each iteration as base models for subsequent iterations as long as there is reduction in AICc (R function in R-Code S1). The fitting procedure constitutes an extension of a more restrictive routine that only included the top ranked model for each iteration in subsequent iterations (Schuster & Arcese, 2013). We opted for the iterative approach because creating all possible models for approach (iv) and (vi) would have resulted in 245 models each. For the remaining approaches, we created models for each possible combination of predictors. For both ZIP and ZINB we further expanded our model lists using (a) intercept only models for the zero-inflation component, while using predictors in the count component and (b) including the same predictors for both zero-inflation and count components. In post-processing we reduced the candidate set of models from each approach based on the statistical significance of all predictors, using p-values as a general and liberal criterion for retaining models. We selected a cutoff value of p = 0.15 as it serves as the default for many stepwise model selection approaches (e.g., Rawlings, Pantula & Dickey, 1998). We chose this approach to reduce the probability of including non-informative models (i.e., those stuck at local maxima for parameter estimation) in subsequent model averaging. Additionally we checked for and removed models with unrealistically high (>50) parameter estimates and/or Standard Errors, which would indicate lack of model fit. For model selection, we ranked all remaining candidate models by AICc and averaged those with ΔAICc ≤ 2 from the top ranked model (Burnham & Anderson, 2002). All analysis were conducted using R v.2.15.2 (R Development Core Team, 2012); package MuMIn 1.8.0 (Barton, 2012) was used for AICc calculations; package MASS v. 7.3-22 (Venables & Ripley, 2002) for NB models; and packages pscl and Formula v.1.1-0 (Zeileis & Croissant, 2010) for ZIP and ZINB models.

To determine the distribution and scale that provided the best relative fit to the data for each mammal group we compared and ranked the models with the lowest AICc of each approach and determined the approach resulting in the overall lowest AICc value. We further contrasted these results with the initial results from the intercept only models to determine whether initial models were sufficient to identify the error distributions that were most appropriate for a given data set or whether predictor variables had to be included first.

Predictive maps of preferred crossing sites

Using the previous model results we created predictive maps for each species/group for both the abundance of animals approaching the highway (transect models) and the abundance of animals, having reached the highway, crossing it (highway models). For each map we chose the approach with the lowest AICc values out of the 4 remote sensing-derived frameworks (200 m, 500 m, 1 km, 3 scales), as landscape level data was not available for the perceptual area polygons. For predictive polygons, we used 30 × 30 m polygons to follow the EOSD resolution. For the highway predictions we created polygons around the highway (line feature buffered 15 m on each side of highway) resulting in predictions for 7374 polygons. For transect predictions we expanded the buffer around the highway to 1 km, resulting in 213442 polygons. Next, we generated a predictor set for each polygon centroid that was identical to those used for survey points, and then estimated abundance based on our averaged models for each of the focal species groups. We combined individual mammal group abundance estimates into 10 quantiles to consolidate focal species maps into an index of site preference, multiplied those scores for each polygon and standardized them by dividing by 1000, resulting in community site preference scores between 0 (being the lowest preference) and 10 (highest preference).

Results

We conducted surveys for 737 km of highway (H) and 118.5 km of transects (T), that yielded the following number of track counts: deer = 970 H/887 T, elk = 575 H/152 T, moose = 65 H/59 T, coyote = 58 H/111 T, bobcat = 6 H/11 T, cougar = 1 H/11 T, wolf = 0 H/10 T, fox = 3 H/2 T. No tracks were found for lynx, marten or wolverine (raw data and R model input files can be found in Data S1).

Road barrier effect

Highway permeability values for the majority of groups were extremely low (where a value of 1 indicates full permeability across the highway and 0 represents no permeability), indicating that Hwy 3 likely acts as a barrier to mammal movement (Table 2). The permeability values for carnivores were only one third those of ungulates (moose, elk, deer, combined) on the investigated section of Hwy 3 (Table 2), indicating that deer, elk, and moose were much less affected by the highway in terms of movement than carnivores. Track accumulation curves for all mammal groups indicate that in all four cases there were areas of the highway where the focal group rarely or almost never crosses the highway (Fig. 2).

Table 2 Permeability values for track counts of the highway and transects along Hwy 3.

Values are given for the community, ungulate and carnivore group levels as well as individual species for all tracks and individual crossings observed. A permeability value of 1.0 indicates no difference between off-road areas and the highway in terms of animal movement.

	All species	Ungulates	Carnivores	Deer	Elk	Moose	
All tracks	0.284	0.307	0.106	0.223	0.895	0.263	
Successful crossings	0.265	0.285	0.104	0.210	0.827	0.221	
	Bobcat	Cougar	Coyote	Fox	Wolf		
All tracks	0.123	0.019	0.121	0.286	0		
Successful crossings	0.123	0.019	0.118	0.286	0		

Figure 2 Cumulative track plots of successful crossing attempts by the four focal species groups.

Areas of no increase indicate locations along the highway where the focal group rarely or never cross the highway. This shows that for some of the focal groups there is substantial stretches of highway that represent crossing barriers.

Landscape variable preference models

Based on intercept only model comparisons using likelihood ratio tests, Vuong test and AIC ranking, the best supported distributions for each mammal group were: ZINB (Deer-Hwy, Elk-Hwy, Deer-Trans), NB (Moose-Hwy, Elk-Trans, Carnivora-Trans), ZIP (Carnivora-Hwy), P (Moose-Trans), indicating that ZINB and NB were the most commonly supported distributions (Table 3). In 6 out of 8 cases, the modeling approach which included predictors from all three scales and the digitized polygons was selected as the top model based on AICc (Table 4). In only two cases for the transect data did other approaches result in lower AICc values: Deer (1 km scale) and Moose (500 m scale). In direct comparison between predictors derived from remotely sensed data and hand digitized data, the remotely sensed model framework resulted in lower AICc values in all eight cases. When comparing the remotely sensed data approach at the same scale as the digitized (200 m) data, digitized predictors resulted in lower AICc values in 7/8 cases. A comparison of the extended modeling results (Table 4) with the initial distribution tests (Table 3) indicates that in 4/8 cases the results from the initial tests were rejected and different distributions formed the basis of models with the lowest AICc values. For Highway and Transect data from each mammal group, model coefficients describing preferred (positive values) and avoided (negative values) habitat variables from the best supported model (above) are depicted in Tables S1 and S2. Summed values of preferred and avoided landscape variables for the entire mammal community are presented in Table 5.

Table 3 Null model comparisons.

Results (likelihood ratio, AICc, and Vuong tests) for initial distribution tests comparing Poisson (Poiss), negative binomial (NB), zero-inflated Poisson (ZIP), and zero-inflated negative binomial (ZINB) distributions on highway and transect abundance for all four mammal groups. Bold values represent the lowest AIC for each comparison. Likelihood ratio tests were performed between Poiss and NB (as well as their respective zero inflated equivalents). Vuong tests were performed between Poiss and ZIP as well as NB and ZINB (p-values for both likelihood ratio and Vuong tests are shown in parentheses.)

		True zeros		logLik	Df	AICc	Pred zero	Likelihood ratio	Vuong	
Carnivores	Hwy	404	Poiss	−204.677	1	411.354	400			
NB	−203.204	2	410.407	404	2.947 (0.086)		
ZIP	−202.762	2	409.524	404		0.995 (0.160)	
ZINB	−202.762	3	411.524	404	3e−04 (0.987)	1.156 (0.124)	
Trans	211	Poiss	−300.184	1	602.367	193			
NB	−282.549	2	569.097	213	35.27 (2.87e−09)		
ZIP	−290.35	2	584.700	211		1.510 (0.066)	
ZINB	−282.549	3	571.097	213	15.603 (7.81e−05)	−1.737 (0.041)	
Deer	Hwy	181	Poiss	−1114.22	1	2230.438	57			
NB	−899.058	2	1802.115	168	430.32 (< 2.2e−16)		
ZIP	−933.312	2	1870.624	181		7.654 (9.66e−15)	
ZINB	−889.12	3	1784.240	181	88.384 (2.2e−16)	2.455 (0.007)	
Trans	110	Poiss	−944.375	1	1890.751	16			
NB	−681.014	2	1366.028	100	526.72 (2.2e−16)		
ZIP	−744.008	2	1492.015	110		7.421 (5.79e−14)	
ZINB	−672.803	3	1351.606	110	142.41 (2.2e−16)	2.211 (0.013)	
Elk	Hwy	181	Poiss	−975.888	1	1953.777	134			
NB	−649.758	2	1303.517	297	652.26 (2.2e−16)		
ZIP	−650.435	2	1304.870	305		9.365 (< 2.2e−16)	
ZINB	−628.978	3	1263.957	305	42.913 (5.724e−11)	3.293 (0.0001)	
Trans	241	Poiss	−350.635	1	703.269	188			
NB	−266.262	2	536.524	240	168.74 (2.2e−16)		
ZIP	−271.467	2	546.933	241		4.356 (6.63e−06)	
ZINB	−265.601	3	537.202	241	11.731 (0.001)	0.5704 (0.284)	
Moose	Hwy	412	Poiss	−203.825	1	409.650	402			
NB	−194.471	2	392.942	412	18.708 (1.524e−05)		
ZIP	−194.774	2	393.548	412		1.652 (0.049)	
ZINB	−194.446	3	394.892	412	0.655 (0.418)	0.120 (0.452)	
Trans	257	Poiss	−162.046	1	326.092	254			
NB	−161.542	2	327.083	257	1.009 (0.315)		
ZIP	−161.253	2	326.506	257		0.662 (0.254)	
ZINB	−161.253	3	328.506	257	3e−04 (0.987)	0.866 (0.193)	

Table 4 Landscape variable preference model results.

Top ranked model AICc values from all model approaches used to determine landscape variable preference across six separate spatial approaches (columns) for all four mammal groups. Bold values represent the lowest AICc of the 4 distributions at one scale. Values in grey background represent the lowest AICc overall for a dataset (Hwy, Trans) and species combination. Values with an asterisk represent the approach used for creating predictive abundance maps for a dataset – species combination.

			200 m	500 m	1 km	3 scales	Digitized	Combined	
Carnivora	Hwy	Poiss	406.72	393.99	393.44	393.44	398.10	384.89	
NB	406.56	394.89	393.98	393.98	398.66	386.39	
ZIP	405.82	394.26	384.43	378.92*	398.14	378.96	
ZINB	407.86	396.29	386.55	381.03	400.20	374.81	
Trans	Poiss	581.72	584.91	584.22	563.26	570.09	546.26	
NB	558.44	559.21	558.78	547.44*	551.32	541.26	
ZIP	566.66	574.32	566.97	557.96	561.51	546.16	
ZINB	560.53	561.29	557.56	571.18	553.44	571.18	
Deer	Hwy	Poiss	2112.12	2113.99	2105.97	2058.87	2153.90	2046.83	
NB	1768.82	1770.12	1766.99	1760.42	1787.08	1760.38	
ZIP	1827.11	1812.58	1820.72	1807.55	1796.08	1768.06	
ZINB	1750.03	1739.02	1739.43	1731.62*	1747.77	1707.97	
Trans	Poiss	1628.13	1523.81	1476.85	1417.29	1652.31	1394.78	
NB	1311.95	1269.85	1250.39	1237.15	1303.81	1233.78	
ZIP	1387.70	1321.88	1332.63	1320.64	1379.71	1309.27	
ZINB	1287.61	1233.46	1231.36*	1238.64	1269.17	1235.47	
Elk	Hwy	Poiss	1833.99	1859.39	1848.05	1754.72	1765.00	1648.51	
NB	1286.84	1293.10	1287.29	1279.93	1285.68	1269.56	
ZIP	1264.16	1273.91	1260.17	1241.21	1243.92	1213.37	
ZINB	1238.56	1239.10	1225.10	1225.10*	1235.30	1219.42	
Trans	Poiss	627.19	575.37	569.61	543.48	589.83	496.10	
NB	513.35	493.12	492.24	472.81	499.24	472.84	
ZIP	530.49	496.65	508.39	476.01	510.29	448.41	
ZINB	515.38	493.81	494.36	469.91*	500.70	459.12	
Moose	Hwy	Poiss	396.80	373.96	371.63	353.20	353.53	318.84	
NB	385.73	365.88	363.09	351.03	349.29	320.59	
ZIP	373.22	351.55	342.48	331.93*	344.41	320.95	
ZINB	375.33	354.47	344.59	334.06	351.33	327.95	
Trans	Poiss	303.36	291.82*	295.15	297.75	311.13	297.31	
NB	313.82	293.94	297.25	299.82	313.22	299.38	
ZIP	313.70	293.94	297.25	301.76	307.72	299.22	
ZINB	315.80	295.53	299.35	309.62	309.91	305.06	

Table 5 Summed importance scores of predictor variables.

The table shows how often a variable was included (as positive or negative predictor) in the eight remotely sensed modeling frameworks used to create predictive maps for Carnivora, Deer, Elk and Moose (marked with asterisks in Table 4).

	Highway	Transect	
	Positive	Negative	Positive	Negative	
Water	2	0	2	1	
Exposed	0	2	1	2	
Low shrub	0	0	0	4	
Wetland	0	0	1	3	
Herbecous	1	1	3	4	
Dense conifer forest	0	0	2	2	
Open conifer forest	0	0	1	2	
Open broadleaf forest	0	2	4	1	
Gravel road length	1	0	5	0	
Paved road length	3	1	1	2	
Number of buildings	0	3	1	1	

Predictive maps of preferred crossing sites

To map preferred crossing sites, we used averaged model results for each mammal group based on the framework with the lowest AICc value out of the 4 remotely sensed model sets (Cells marked with an asterisk in Table 4; maps and shapefiles in Figs. S1–S4 and Data S2 respectively). The combined predictions of preferred (green) and avoided (red) crossing sites for all investigated species within the mammal community are illustrated in Fig. 3. Based on predictions generated from landscape variables (above), certain regions of the study area exhibited high preference scores from both approach (transect) and crossing (highway) models (e.g., Fig. 3 insert A), indicating that these locations likely represent areas of high priority when implementing mitigation measures for all species considered in our study. Conversely, certain regions of the study area exhibited high preference scores for one of the model sets (crossing vs. approach), but not the other (e.g., Fig. 2 insert B), indicating that these may represent less-ideal locations to implement mitigation measures such as crossing structures. Areas of unambiguous preference for particular crossing sites (i.e., those where crossing and approach preference scores overlap) differ for each mammal group considered in our study (Figs. S1–S4), indicating that mitigation strategies aimed at mammal communities may differ substantially from those aimed at a target species.

Figure 3 Community crossing site preference (green) and avoidance (red) for highway approach and actual crossing predictions.

Crossing predictions are visible in inserts A and B as the polygons in the center within the highway outline. Results are based on averaged model results from the best remote sensed model framework for the carnivore group, deer, elk and moose. Individual model framework abundance predictions were split into 10 quantiles, multiplicatively combined and standardized by dividing by 1000 to create community scores between 0 and 10. None of our predictions approach the maximum of 10 as no location suits all species perfectly. Insert A shows an area with high overlap between approach and crossing scores. Insert B illustrates and area of high crossing scores but low approach scores.

Discussion

We determined that Hwy 3 posed a severe movement barrier to the local mammal community. Although each investigated species differed in the landscape variables associated with preferred and avoided crossing sites, we used a multi-scale approach to identify locations along the highway where mitigation measures may benefit all species in the large mammal community. Below we address our earlier questions and discuss the implications of our finding that multi-scale habitat assessments may be necessary to accurately predict the most effective locations for highway crossing structures (e.g., culverts and overpasses) or other mitigation measures.

Permeability estimates for both carnivores and the majority of ungulate species considered were extremely low across the highway (Table 2), indicating that Hwy 3 likely acts as barrier to animal movement. Although permeability estimates for elk were comparatively high (likely due to herding behavior, whereas tracks for all other species tended to be solitary or in small groups), averaged estimates for all ungulates and the entire mammal community suggest that movement by large-bodied mammals is highly restricted across the highway. Likewise, track accumulation curves (Fig. 2) indicate that for each species group considered, certain areas of the highway may rarely or never be crossed, posing large limitations to population connectivity across Hwy 3. This finding is consistent with previous estimates of wildlife permeability across a similar highway through the Rocky Mountain Range of Alberta, Canada (Alexander, Waters & Paquet, 2005). Such low permeability across the highway suggests a severe threat of habitat fragmentation to the mammal community, which could result in decreased gene flow across the road barrier, and ultimately to lower population viability in the region (Mader, 1984; Epps et al., 2005). These results indicate a need to accurately identify locations for potential mitigation measures along roads such as Hwy 3 to facilitate the movement of individuals across the highway and reduce this barrier effect (Harrison & Bruna, 1999; Haddad et al., 2003; Crooks & Sanjayan, 2006).

By incorporating both highway and transect predictions simultaneously, we aimed to identify locations for potential mitigation measures that represent both preferred crossing sites as well as preferred approach habitat up to 1 km from the highway. We determined that the landscape variables associated with preferred/avoided crossing sites differed for many of the mammal groups considered (Tables S1 and S2). In all cases, noise generated from vehicles travelling on the highway could contribute to road avoidance by large mammals (Forman & Alexander, 1998; Jaeger et al., 2005; Barber, Crooks & Fristrup, 2010). However, numerous studies on movement across roads by large and small mammals have found no consistent response to noise levels, and suggest that habitat characteristics surrounding crossing sites play a larger role in animal movement than individual tolerance to noise levels (McGregor, Bender & Fahrig, 2008; Iglesias, Mata & Malo, 2012). For instance, carnivores tended to avoid residential areas along the highway as well as open areas with low shrub cover (Tables S1 and S2), consistent with previous studies (e.g., Mech, 1995). While elk and deer did not avoid these landscape features, these two species exhibited dissimilar patterns of habitat and crossing-site preference, consistent with their different habitat requirements (Johnson et al., 2000). These differing results per group indicate that a clear set of conservation goals for each species as well as the community as a whole must be established before mitigation measures are implemented to facilitate highway crossing (e.g., Beier, Majka & Spencer, 2008).

We used multi-model inference and model averaging to identify locations of preferred crossing sites for all mammal species considered, which would likely serve as the most effective locations for mitigation measures aimed at increasing mammal permeability across the highway. Cumulative scores of preferred/avoided landscape variables along both the highway and transect data sets indicate that preferred crossing sites tended to be within close proximity of water and longer stretches of unpaved road (Table 5). Crossing-specific scores indicate a preference for longer stretches of paved roads, and approach-specific scores suggest preference for areas of high crown cover with abundant broadleaf trees, respectively. Although this approach may reduce the efficiency of predicting highway crossing sites for certain focal species, community-level approaches are increasingly advocated as a more efficient means of implementing wildlife linkages across barriers such as major roads (Beier, Majka & Spencer, 2008). To accomplish this goal, we applied an exhaustive model approach incorporating four separate distributions of abundance for each mammal group along Hwy 3. In only 4 of the 8 cases considered was pre-selection of the y-distribution successful, indicating that an exhaustive modeling approach incorporating multiple distributions may be necessary when the goal is to identify and predict preferred crossing sites based on limited data and uncertainties regarding which abundance distributions are most applicable to free-living animal populations. By adopting the approach described here, researchers may be able to extract more information from highway crossing data than could otherwise be gained from applying predefined and potentially inaccurate abundance distributions. Further, the best-supported distribution differed for each species; while ZINB and NB were the most commonly supported distributions, NB, ZIP and P each received the best support for at least one data set (highway versus transect). These results once again highlight the need for future studies to consider the unique habitat requirements of each species within mammal communities when developing mitigation strategies, but that those strategies which provide the greatest benefit to the largest number of species should be given priority for implementation.

To establish conservation-based goals for large mammals along roads such as Hwy 3, further consideration must be given to whether the spatial scales at which habitat characteristics are measured match the spatial scales at which the animals select preferred/avoided crossing sites. We determined that in 6/8 cases, a combined approach to modeling preferred crossing sites (incorporating remotely sensed and hand-digitized predictors) resulted in the best supported model. Further, utilizing multi-scale remote sensing-derived predictors always resulted in better model support than utilizing only hand-digitized predictors for each species and data set considered. Thus, our results indicate that while a combined approach may represent the most informative method for predicting landscape variables of preferred mammal crossing sites, freely-available macro-habitat data such as those generated through remote sensing may be more useful than labor-intensive micro-habitat assessments when time and budgetary constraints on data collection are imposed. Previous studies investigating habitat occupancy in birds have found similar results (e.g., McClure, Rolek & Hill, 2012; Meiman et al., 2012), highlighting the increasing usefulness of remote sensing in evaluating localized questions in conservation and community ecology.

The goal of our study was to identify locations along Hwy 3 where mitigation measures might increase connectivity across the highway for all species in the mammal community. Although we do not currently have data on which mitigation measures may be the most effective on increasing permeability in this system, previous studies investigating the costs/benefits of different mitigation strategies at the community level (e.g., Clevenger & Waltho, 2000; Clevenger & Waltho, 2005) indicate that a diversity of crossing structures of different sizes may best serve large mammal communities. Because our permeability estimates were based on snow tracks and not on data for the entire year, there is the potential for our results to only be applicable for winter months. Further, because our permeability estimates are based on transects with a mean distance of 175 m from the highway, we likely overestimate permeability in certain cases by not considering the density of animals in areas further away from the highway. For instance, Dickson & Beier (2002) determined that cougars typically avoid high speed roads at a distance of 500 m–1 km and more generally, mammal populations might be influenced by human infrastructure up to about 5 km (Benítez-López, Alkemade & Verweij, 2010). Although conducting further transects at a greater distance from the road may improve estimates of habitat preference for each species along Hwy 3, we believe our methods represent a realistic investigation of the types of habitat used by animals approaching and ultimately crossing the road, which may help inform strategies for implementing crossing structures. A potential limitation to our approach of determining the most appropriate locations for multi-species crossing structures is that preferred landscape traits differed among groups, indicating that some species would benefit less from crossing sites that serve the majority (for species specific preferences see Figs. S1–S4). While the specifics of which species should be given priority in such an instance will depend on the conservation goals of managers, our method presents a potentially viable way of increasing highway permeability for multiple species, and ultimately improving connectivity and population viability for mammal communities along major roadways.

Although our study was limited to one section of highway, its importance as a wildlife corridor suggests that our approach may be widely applicable to other areas where roads bisect important wildlife habitat. In situations where managers are capable of implementing mitigation measures aimed at increasing cross-road permeability for multiple mammal species, future studies should seek to evaluate the efficiency of this method over traditional single-species approaches. Specifically, to verify the effectiveness of our approach compared to a single-species mitigation strategy, managers would ideally implement our method in areas where traditional mitigation approaches have been in place for a number of years. By directly comparing permeability values before and after the implementation of a multiple-species mitigation approach, we may gain further insight into benefits of community-level conservation planning.

Finally we would like to acknowledge that our modeling approach only constitutes one possible way of drawing inference about highway approach and crossing behavior of the investigated mammal community. Here, we provide a flexible but somewhat restrictive framework for predicting animal abundance. Though there is always uncertainty surrounding model choice when using a multi-scale approach, extra caution should be used when basing model choice on ‘stepwise’ procedures and using p-values to exclude certain models from a set. The use of AIC to rank models is currently widely applied in the literature and is assumed to be valid, but this approach only gives a relative measure of fit for comparing models. AIC does not provide a measure for predictive ability of a model, which should ideally be tested against additional data. Finally, alternatives to model averaging such as a reversible jump MCMC approach (Green, 1995) could be employed to compare results and further improve robustness of analysis.

Conclusion

Roads such as Hwy 3 represent severe barriers to animal movement and pose a major threat to wildlife habitat, but few studies investigate how or where to implement mitigation measures at the community level. We identified areas along the highway with habitat features of preferred crossing sites for eight species of large mammals, representing locations where mitigation measures may have positive effects for all species investigated. We determined that a combined approach incorporating both remotely sensed and hand-digitized landscape variables best predicted crossing site preference for most species, but that remote sensing data was always better than hand-digitized values when utilized separately. Our results indicate that a multi-scale approach may be necessary when identifying areas to implement mitigation strategies across roads, as differing habitat requirements for members of the mammal community may limit the usefulness of single-species, single-scale approaches.

Supplemental Information

Figure S1 Abundance predictions for the Carnivores group based on averaged model results from the best EOSD model framework

Click here for additional data file.

Figure S2 Abundance predictions for Deer based on averaged model results from the best EOSD model framework

Click here for additional data file.

Figure S3 Abundance predictions for Elk based on averaged model results from the best EOSD model framework

Click here for additional data file.

Figure S4 Abundance predictions for Moose based on averaged model results from the best EOSD model framework

Click here for additional data file.

Data S1 Datasets containing raw and modeling data

Data_final contains the collected raw data including a metadata sheet. Hwy_Modelling_Data contains only Highway data including covariates used in the R analysis. Trans_Modelling_data contains only Transect data including covariates used in the R analysis.

Click here for additional data file.

Data S2 Result shapefiles for the Highway and Transect predictions

Included are two ESRI shapefiles, one for the Highway and one for the Transect buffer, containing the predicted abundance estimates for all species/groups based on the averaged model from the best EOSD framework used to create Figs. S1–S4 as well as the community estimates used to create Fig. 2.

Click here for additional data file.

R-Code S1 Iterative model selection R function script

Click here for additional data file.

Table S1 Averaged model coefficients based on the overall best model framework (can including both EOSD and conceptual polygons scales)

Click here for additional data file.

Table S2 Averaged model coefficients based on the EOSD best model framework (used to create the landscape level predictions)

Click here for additional data file.

We thank D Quinn for logistical support throughout data collection, SM Alexander for helpful advice on data collection and analysis, and W Desch for initial methodological and statistical advice. F Suppan, P Beier, J Jenness and K Crooks provided feedback on GIS-analyses, and AE Passmore helped edit a previous version of this manuscript. We thank G Stewart, P Beier, and two anonymous reviewers for comments and suggestions on an earlier draft of this manuscript.

Additional Information and Declarations

Competing Interests

Author Contributions

The authors declare no competing interests.

Richard Schuster conceived and designed the experiments, performed the experiments, analyzed the data, contributed reagents/materials/analysis tools, wrote the paper.

Heinrich Römer contributed reagents/materials/analysis tools, wrote the paper.

Ryan R. Germain analyzed the data, wrote the paper.

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
