# Peer review of "Using multi-scale distribution and movement effects along a montane highway to identify optimal crossing locations for a large-bodied mammal community"

_PeerJ, doi:10.7717/peerj.189_

## Round 0.1 · original submission · Minor Revisions

· Academic Editor

Minor Revisions

The work is a useful contribution, but responding to the reviewers comments will result in improvements. i would encourage you to amend the manuscript to address the points they make where possible particularly in the case of reviewer one. Reviewer three is critical of your methods for model selection. I would suggest that you clarify the approach you adopted in the methods and results and include some discussion of the range of sensitivities. If you wish to add more discussion of the uncertainties surrounding model choice, it might be worth discussing reversible jump MCMC as an alternative to model averaging, perhaps as a suggestion for further work. Highlighting the dangers inherent in using p values or stepwise approaches would also be useful. I do not think that you need to adopt a new strategy, simply explain the potential problems a little more. Finally I would also encourage you to run a web of science search, to make sure that no meta-analyses are available on this topic. yours truly

Gavin Stewart

Reviewer 1 ·

Basic reporting

The article meets all basic reporting standards. It is well written, clear, and appropriate for publication.

Experimental design

I am uncertain on the methodology. I am not familiar with animal tracking or movement ecology to any real extent. However, use of rangefinder cameras or some other sampling methodology to confirm that the method used herein, tracks in show, was effective. What if animals move a lot less when it has snowed? or move very differently? This could have been easily validated as a method and would have made this paper far more useful.

Validity of the findings

Valid findings provided the methodology effectively estimates animal movements.

Additional comments

Good paper. Useful. Please consider linking to theory a bit more with respect to habitat frag, connectivity theory, circuit theory for conservation, or whatever is appropriate. Also, are there any other additional stats to be added to explore impermeability of roads or natural mitigation opportunities that may be available to us to promote movement, ie any covariates in landscape etc.

Please also add #total number of tracks per species etc. I think you just present the crossings per km. Finally, can you apply rarefaction curves or some other measure to explore accumulation of species or tracks?

·

Basic reporting

This paper describes and illustrates methods for identifying optimal locations for highway-crossing structures for a diverse mammal community. It demonstrates the superiority of free, remotely-sensed data to costly hand-digitized data, the superiority of multi-scale models to single scale models, and presents one approach to integrating across single-species models. The methods are appropriate (with one possible exception) and clearly described. Inferences are well-grounded in the data and statistical analyses.
The intro states a major objective that is not mentioned in the Abstract. The abstract should state: “Freely-available remotely-sensed habitat landscape data were better than more costly, manually-digitized microhabitat maps in supporting models that identified good crossing sites; however models using both types of data were better yet.” The Abstract should also mention that in 6 of 8 cases the multi-scale models performed better than models at any single scale.
The paper is generally well-written and clear. I think you should feel free to be a bit more punchy and colloquial. For instance, instead of dryly referring to the “highway and off-highway transect models,” you could refer to the latter as a model of the probability that an animal approaches the highway, and the former as a model that an animal, having reached the highway, crosses it. The title should highlight the idea of identifying optimal locations for crossing structures. And the table captions should go much further in telling the reader “what’s the point.” Right now every caption is “Here are some numbers. Guess why we put them there. Guess what they mean.” See Kroodsma (2000. Auk 117:1081-1083).
Table 1 can be improved. State that the unit of measurement is “% of area” unless otherwise stated. What is the unit of measurement for road/path, and railroad? Replace the acronym column with a “variable name” column that uses clear descriptions (“% unvegetated” instead of “no_veg” and “% dense conifer forest” instead of “con_d”). Reverse the column orders (1Variable name, 2Variable description, 3Source). In the caption or in a table footnote, state that the PAP variables were hand-digitized from 1-m photos (just to remind the reader).
Table 2. Spell out words. If “Ind crossings” means “successful crossings” use that term (spelled out).
Table 3, 4, 5. Captions too terse.
Table 3 Caption should explain the point of the table.
In Table 5, write out the variable names; transpose rows and columns so the names will fit.
You seem to use “covariate” and “parameter” and “predictor” and occasionally “characteristic” interchangeably, for the same things that are more commonly referred to as “explanatory variables” or “variables.” Not a big deal, but I think the terms “variables” and “predictors” are less-offputting and more accessible to most readers. Line 167: “landscape parameter” should probably be “land cover or condition”?

Experimental design

My one real concern is with averaging the maps predicting the best crossing sites for individual species to produce a map intended to serve all species. If a site A is excellent (10 on a scale of 10) for 9 species and 0/10 for the 10th species, the average is 9. If site B scores 8 for all species, its average is 8. But I’d argue that B is better, no? On lines 338-9, you state that landscape traits associated with preferred crossing sites differed among species, suggesting that some species could be poorly served by a site that serves the majority. Please discuss limitations and advantages of averaging as a way of combining predictions across species, and alternative (existing or potential) ways of combining across species. Perhaps a post-hoc procedure to identify “losers” and accommodate their needs.
Other than this one issue, this is well-designed and well-executed. Thank you for acknowledging me for feedback on the GIS analyses, even though I do not recall providing any such feedback. At this point I feel you have more to offer me on this topic than I can offer you.

Validity of the findings

The Discussion might mention that the road permeability calculation almost certainly overestimates road permeability because the denominator (# of crossings of transects 10-900m from highway, mean 175 m from highway) does not reflect the numbers of animals who avoided the road at an average distance > 175 m.
In the abstract, I suggest deleting “likely resulting in population fragmentation” unless you present evidence this is true. Puma data are too sparse to support inference about fragmentation. For other species, perhaps 10% passage rates are enough to avert fragmentation.

Additional comments

Thank you for the opportunity to review this fine contribution.

Reviewer 3 ·

Basic reporting

The subject, methods, and results for this paper are presented clearly. The paper is very well written with only a few typographical errors. Some components of the methods, in particular sample design, could be more clearly explained. I make specific comments below for the authors.

Experimental design

I had no concerns with the substantive components of the paper. The data and methods fit the objectives very nicely. Results were followed by a lucid discussion of the findings. My primary concern is the choice of method for model selection and validation.

The authors conduct an exhaustive analysis of the 4 types of count distribution. Interesting, but often these details are subsumed in the unreported elements of an analysis.

The hybrid information theoretic(IT)/p-value approach for identifying the most ‘parsimonious’ model was overly complex, obtuse and technically flawed. The authors selected/combined models using a combination of best subsets, screening based on the p-values of the covariates, and finally model averaging. The best subsets method can result in models that are particular to a set of data – this is useful for exploring data, but not developing models that generalise to other study areas or time periods. Choosing individual variables based on their p-values assumes that the coefficient and SE are independent of other variables in the model. This is not the case – each covariate is dependent on the contribution of other covariates. Furthermore, this mixes two model selection philosophies: IT and hypothesis testing. I don’t know if a simpler model selection process (i.e., identify 15-20 models based on existing literature and theory and select best model using AICc delta) would change the results, but the current approach is certainly awkward if not statistically incorrect.

AIC provides a relative not absolute measure of model fit. Thus, we have some idea of what is the best model of the set, but not if that model has any ecological or predictive validity (i.e., even the best model could be a very poor predictor). The best model should be tested against a set of withheld data; this might involve testing the observed versus predicted probability for each count or by looking at the simple residuals (observed minus predicted counts for the withheld data). This is especially important when considering that these models will be used to predict counts on the highway and guide the location of mitigation strategies.

Validity of the findings

Although the model selection process was odd and perhaps incorrect (see previous section), the sampling and data are valid and the general statistical treatment was correct. The results support the objectives and conclusions of this work.

Additional comments

General Comments
The subject, methods, and results for this paper are presented clearly. The paper is very well written with only a few typographical errors. Some components of the methods, in particular sample design, could be more clearly explained. I make specific comments below for the authors.

I had no concerns with the substantive components of the paper. The data and methods fit the objectives very nicely. Results were followed by a lucid discussion of the findings. My primary concern is the choice of method for model selection and validation.

The authors conduct an exhaustive analysis of the 4 types of count distribution. Interesting, but often these details are subsumed in the unreported elements of an analysis.

The hybrid information theoretic/p-value approach for identifying the most ‘parsimonious’ model was overly complex, obtuse and technically flawed. The authors selected/combined models using a combination of best subsets, screening based on the p-values of the covariates, and finally model averaging. The best subsets method can result in models that are particular to a set of data – this is useful for exploring data, but not developing models that generalise to other study areas or time periods. Choosing individual variables based on their p-values assumes that the coefficient and SE are independent of other variables in the model. This is not the case – each covariate is dependent on the contribution of other covariates. Furthermore, this mixes two model selection philosophies: IT and hypothesis testing. I don’t know if a simpler model selection process (i.e., identify 15-20 models based on existing literature and theory and select best model using AICc delta) would change the results, but the current approach is certainly awkward if not statistically incorrect.

AIC provides a relative not absolute measure of model fit. Thus, we have some idea of what is the best model of the set, but not if that model has any ecological or predictive validity (i.e., even the best model could be a very poor predictor). The best model should be tested against a set of withheld data; this might involve testing the observed versus predicted probability for each count or by looking at the simple residuals (observed minus predicted counts for the withheld data). This is especially important when considering that these models will be used to predict counts on the highway and guide the location of mitigation strategies.

Detailed Comments
Abstract
The abstract is well written and provides a balanced summary of the key objectives and results of the paper. However, some additional description of results (1 additional sentence) would be useful.
L6: comma should follow “Canada”

Introduction
L57-59: This sentence is somewhat confusing and should be simplified – doesn’t modelling animal movement result in predictive models?
L60: Unclear why there is an emphasis on migratory birds and reptiles; there is much published evidence of multi-scale habitat selection and movement by mammals as well.
L77-84: The authors provide a very nice statement of the study objectives.

Methods
L104: The number and type of sampling locations should be a simple idea, but it is not clearly explained. In particular, why were transects surveyed versus the edge of a road and how do these 10 transects relate to the 9 transects on L128? Why is there a reference to a right-of-way – is this the highway or another type of linear feature? Did the authors survey highways and off-road/gravel roads?
L108-109: The methods for sampling tracks/transects is key to understanding the results and should be reported not referenced.
L202-208: Not necessary to derive the Vuong test; a citation will suffice.
L217-231: This is an overly complex approach for model selection that some would classify as data dredging. Using a classic IT approach, each model should serve as a hypothesis. The approach used by the authors might result in a very predictive model, but the best subsets approach will likely capitalise on unique correlations that limit the generalisation of the ‘best model’. Furthermore, the method of selecting variables based on individual p-values is unusual and fails to consider the covariation of the covariates (i.e., the magnitude and variance of any one covariate is a function of other covariates in the model, thus, they cannot be considered as unique entities).
L237: AIC provides a relative not absolute measure of model fit. See comment above.
L239: What is meant by “exhaustive model tests”?
L248: The identity/source of the polygon is unclear; is a polygon a pixel? If so, then just refer to pixels.
L253: Are the “mammal group estimates”, estimates of the counts of crossings?

Results
L265-266: I recommend presenting tables in parentheses only; no reason to waste text to introduce a table.
L279: Did the authors screen for excessively high multicollinearity using VIF/tolerance scores? The inclusion of the same variable at the three scales suggests that this might be a problem.

Discussion
The discussion was well formulated. However, there is much reference to mitigation strategies, but no mention of what those strategies might be. Two or three sentences describing mitigation would be of value to those readers not familiar with highway planning.

Tables and Figures
Table 5: Not clear what is meant by “selection frequencies”; are these the averaged coefficients from the best models?
The figure captions are too brief – they should state where and what, relative to the objectives of the study.

Figure 2: how does one differentiate between the predictive scores for the transect and the highway?

---

## Round 0.2 · accepted · Accept

· Academic Editor

Accept

Thank for you for clarifying the points raised by myself and the reviewers, and for constructive engagement with peer review. I think this paper is a great addition to the literature.